# Physical Activity Levels during Therapeutic Camp Activities in Youth with Disabilities in the United States

Vincenzo G. Nocera [1], Tyler J. Kybartas [2], Angela J. Wozencroft [3] and Dawn P. Coe [3,*]

1   Department of Health and Human Performance, Plymouth State University, Plymouth, NH 03264, USA
2   School of Kinesiology and Recreation, Illinois State University, Normal, IL 61790, USA
3   Department of Kinesiology, Recreation, and Sport Studies, The University of Tennessee, Knoxville, TN 37996, USA
*   Correspondence: dcoe@utk.edu; Tel.: +1-865-974-0294

**Abstract:** Youth with developmental disabilities (DD) face challenges that may impact their participation in physical activity. One of the biggest challenges is the availability of opportunities to engage in activities that are adapted for youth with DD. In addition, due to challenges with current physical activity assessment methods for youth with DD, the activity levels during modified activities remain unclear. The purpose of this investigation was to determine the activity levels of youth with DD during structured and unstructured activities offered during a therapeutic camp. This camp was a five-day, overnight experience in an outdoor camp center in the southeastern region of the U.S. Youth ($n = 29$; $14.6 \pm 3.9$ years) with more than one DD and with varying abilities wore accelerometers while they engaged in 13 activities of varying categories (functional/gross motor, game, sociodramatic, fine motor, free play) and contexts (ropes, horses, outdoor adventure, music and movement, yoga, come on down, sports and games, theatre, cabin challenge, arts, cooking, mad science, free play). Activity level varied by activity category and context and the intensity level of the majority of the camp activities was classified as either sedentary or light. There was a time course effect on activity; most activities resulted in a gradual decline over the session, except for cooking, sports and games, and free play. This therapeutic camp provided an opportunity for youth to engage in physical activity that would be classified as light intensity. The activities available at this camp were designed to address specific goals and objectives and provided enrichment opportunities (e.g., life skills, social skills) for youth to obtain multiple skills while using movement as a framework to deliver the content.

**Keywords:** children; adolescents; play; accelerometry; recreation



## 1. Introduction

In general, physical activity has been shown to positively impact quality of life [1]. Benefits of physical activity include reduced risk for premature mortality, cardiovascular disease, hypertension, obesity, some forms of cancer, depression, and improved cognitive function [2] in individuals with and without disabilities. For all children and adolescents, these benefits have been shown to occur with at least 60 min of daily moderate to vigorous physical activity (MVPA). Therefore, it is recommended that all youth accumulate 60 min of MVPA daily that is developmentally appropriate [2]. Approximately 15% of youth of ages 3 to 17 years have developmental disabilities, which include conditions that result from mental and/or physical impairment [3]. Preliminary evidence suggests that youth with developmental disabilities participate in lower levels of MVPA compared to their peers [4]. Therefore, if youth are not meeting physical activity guidelines [2], it is critical that they engage in some type of movement in order to reduce time spent in sedentary pursuits (i.e., screen time, TV watching). The Canadian 24-Hour Movement Guidelines for Children and Youth includes recommendations for sedentary behavior ($\leq 2$ h of daily screen time, limit sitting for long periods of time) and stepping (light intensity physical activity) in addition to 60 min of MVPA per day [5,6]. The World Health Organization (WHO) recently

released physical activity guidelines for individuals with disabilities. The WHO offers similar recommendations for MVPA and sedentary behavior/screen time [7]. Additionally, these guidelines state that all physical activity counts, with health benefits achieved even when youth are not meeting recommendations. This document includes a statement that "doing some physical activity is better than doing none" [7]. It is important to recognize that many children and adolescents with developmental disabilities are not provided opportunities to be physically active due to a variety of reasons. These reasons include a lack of programs developed and adapted for children with developmental disabilities [8], a deficiency of adapted equipment [9], and inadequate training of staff members who provide recreational physical activity [10]. Therapeutic camps provide opportunities for youth with developmental disabilities to participate in leisure and recreational activities and subsequently accumulate physical activity. Typically, therapeutic camps offer higher staff to camper ratios with specific training to meet the needs of the campers either through a day or residential program. Such camps provide both medical care and therapeutic activities focusing on specific goals and objectives to address campers' limitations and skill development. Therapeutic camps have been shown to have numerous benefits including increased social self-efficacy and performance [11]. Additionally, for individuals with developmental disabilities, these camps may contribute to positive youth development [12]. Lastly, increased enjoyment, interest, relaxation, and intrinsic motivation have also been reported as benefits associated with therapeutic camps [13,14].

Although, several benefits of therapeutic camps have been reported, the amount and patterns of physical activity accumulated during these programs remains unclear. The goal of therapeutic recreation camps is to participate in activities tailored to the abilities and needs of the participants. While physical activity enhancement may not be the primary objective of camps, the environment and programming are conducive to participating in structured and unstructured physical activities. For example, structured activity allows individuals to participate in pre-planned or instructor-led activities that may contribute to the development of motor competence and confidence for a variety of skills [15] Furthermore, structured activities appear to foster physical activity participation, specifically for those with autism spectrum disorder [16]. Conversely, unstructured activities provide the opportunity for individuals to freely choose activities that interest them. This allows youth to engage and experiment with different types of movements, in a spontaneous manner [17]. Camp provides an autonomous setting to investigate physical activity levels during a variety of activities in a diverse population. Camp also provides opportunities to expose youth to a variety of activities that can also be performed at home, school, or in a community setting.

Thus, a stronger understanding of physical activity levels in youth with developmental disabilities during different types of physical activities may ultimately contribute to developing appropriate intervention programs for this population. There is a critical need for researchers and practitioners to better understand the types of physical activity that most effectively lead to the accumulation of activity of all intensities for youth with developmental disabilities. However, it is important to note that the method of measurement is crucial to evaluating physical activity in settings such as therapeutic camps. Accelerometry provides an objective measure, with moderate levels of feasibility and validity that make these devices the most used in both research and in evaluation of physical activity [18–21]. Objective measures of physical activity provide data with limited bias, whereas bias is typically found in methods such as surveys and questionnaires [22], especially when working with individuals with some form of cognitive impairment. The use of accelerometers and related data can provide useful feedback regarding physical activity levels and patterns to help establish programs that can be used to design or improve programing for youth with disabilities [23]. Accelerometers have the ability to measure overall physical activity levels and can discriminate among different intensity levels of activity [22,23]. This methodology is the primary assessment tool used to measure physical activity in studies such as the National Health and Nutrition Examination Study (NHANES; United States) and the

Biobank Study (United Kingdom), as accelerometers are considered accurate and feasible for use in diverse populations [24,25]. Additionally, Chiew et al. reported that wearable devices, such as accelerometers, complement other assistive technologies for persons with developmental disabilities [26]. Accelerometers have integrated technology that is similar to the mechanisms used in consumer wearables such as Apple watch, FitBit, etc. In addition, since accelerometers resemble wearable technology, this may increase the likelihood that youth will wear the devices during research studies and in the evaluation of programs. Therefore, the purpose of this investigation was to determine the physical activity levels and intensity of activity of youth with more than one developmental disability during 13 activities (structured and unstructured) offered during a five-day overnight therapeutic camp through the use of wearable accelerometer technology.

## 2. Materials and Methods

### 2.1. Study Participants

A total of 29 campers (aged 7–21 years), with developmental disabilities volunteered to participate in this study. Campers could be enrolled in the study if they met the following inclusion criteria: camp attendee and walks with a typical gait, with hand crutches or walker, or with ankle/foot orthoses. There were no exclusion criteria. Descriptive characteristics are presented in Table 1 and the list of participants' primary developmental disabilities is included in Table 2. All participants were attending a five-day, overnight therapeutic camp in the southeast region of the U.S. Prior to data collection, an email detailing the study's purpose and methods was sent to the parents and/or guardians of those attending camp. After check-in, informed consent was obtained from parents and/or guardians. During this time, study protocols were thoroughly explained to parents and/or guardians and to the study participants. Additionally, participants were asked to provide written assent prior to their participation. This study was approved by the Institutional Review Board at the affiliated university.

**Table 1.** Descriptive characteristics ($n$ = 29; mean $\pm$ standard deviation).

| | |
|---|---|
| Age | 14.6 $\pm$ 3.9 |
| Mass (kg) | 59.3 $\pm$ 24.6 |
| Height (cm) | 152.8 $\pm$ 18.0 |
| BMI (kg/m$^2$) | 24.5 $\pm$ 6.7 |
| Sex | |
|     Male | 45% |
|     Female | 55% |

**Table 2.** Primary disability of each participant ($n$ = 29).

| | |
|---|---|
| Autism ($n$ = 6) | Genetic disorder ($n$ = 1) |
| Cerebral Palsy ($n$ = 1) | Global Apraxia ($n$ = 1) |
| Developmental Delay ($n$ = 1) | Global Developmental Delays ($n$ = 1) |
| Disgenisis of the Corpus Callosum ($n$ = 1) | Intellectual Disability ($n$ = 4) |
| Down Syndrome ($n$ = 9) | Prader–Willi Syndrome ($n$ = 1) |
| Dup15q Syndrome ($n$ = 1) | Spina Bifida ($n$ = 1) |
| Tetrasomy X ($n$ = 1) | |

### Camp Setting

Data were collected as part of a therapeutic camp in the spring of 2018, at an accessible 4-H camp located in the southeast region of the U.S. The purpose of this therapeutic camp was to provide individuals with developmental disabilities the opportunity to participate in an outdoor camp that offers an experience that would be similar to traditional camping activities plus additional enrichment activity opportunities. These enrichment activity opportunities (e.g., life skills, social skills) are designed for youth to obtain and develop multiple skills while using movement as a framework to deliver the content. At this specific

camp, each camper was assigned at least one counselor, resulting in a 1-to-1 camper-to-counselor ratio.

Each camper was placed into a cabin group based on age and sex. Cabin groups participated in 14 activities of varying contexts based on the primary activity skill addressed, for a total of 41 min per activity in station-like format. The activities were planned and implemented by university students following the approval of a Certified Therapeutic Recreation Specialist. Activities were offered in a round robin schedule for each cabin group, with nine activities on the schedule for each day. The campers engaged in each activity at least two times during the camp. Activities were named based on the context of each activity. The activities included ropes, outdoor adventure, horses, music and movement, come on down (game show activities), sports and games, theatre, cabin challenge (inter-cabin competitive task), arts, cooking, mad science, yoga, free period, and canoes. All activities were assigned to their activity category based on the major objective of that activity: gross/functional, game, sociodramatic, fine/constructive, and free play. The activity categories were agreed upon by all authors. A detailed description of each activity and how it was categorized is presented in Table 3. With the exception of the gross/functional and free play, the majority of the activity in the remaining categories was primarily performed in the stationary position (sitting or standing). For safety reasons during the canoeing activity, the counselors were responsible for paddling the canoe and the camper was instructed to remain still and limit movement. Due to this lack of direct camper involvement and no physical activity in the canoes, this activity was not included in the analysis.

**Table 3.** Camp activity descriptions and classification table.

| Activity | Description |
| --- | --- |
| **Gross/Functional** | |
| Ropes | Movement at a self-selected pace through a tight rope like obstacle course. Movement would primarily be classified as balancing and walking. |
| Horses | Engagement in actual horseback riding with assistance. Activity would be limited to upper body movement with the lower body providing stability on the horses. |
| Outdoor Adventure | Learn and execute basic survival skills such as movement though obstacle courses at a self-selected pace, typically walking. |
| Music and Movement | Learn and execute dance movements. |
| Yoga | Learn and execute basic yoga movements. |
| Canoeing * | Ride in a canoe while counselors paddle. Movement is restricted during this activity for safety reasons. |
| **Game** | |
| Come on Down | An interactive game that tests the knowledge of popular culture in a game show format, participants alternate between seated and standing positions. |
| Sports and Games | A variety of sports and games played within a controlled and inclusive environment. |
| **Sociodramatic** | |
| Theatre | Development and performance of pretend play on stage. The participants alternated between seated and standing positions. |
| Cabin Challenge | Challenging tasks that stimulate cognitive functioning. Participants were seated for the majority of this activity. |
| **Fine/Constructive** | |
| Arts | Consisting of various artistic projects. Participants were seated for the majority of this activity. |
| Cooking | Learn and execute basic cooking skills and techniques. Participants were seated for the majority of this activity. |
| Mad Science | Participate in a variety of scientific experiments. Participants were seated for the majority of this activity. |
| **Free Period** | |
| Unstructured free time | A free period of time with no planned activities. Activities varied during this time. |

\* Due to lack of participant involvement and physical activity while canoeing, canoe was not included in the analysis.

### 2.2. Study Design

Prior to data collection, the participants and their counselors met with the researchers. The researchers introduced themselves and familiarized the youth with the accelerometers. During this time, height was assessed using a portable stadiometer and weight was measured using an electronic scale. In order to introduce the accelerometer to the participant and to ensure optimal compliance, all individuals had the opportunity to decorate their own accelerometer band. This was done prior to the commencement of camp activities. Additionally, various staff members were affixed with their own accelerometer, this was done to not only ensure that campers were wearing the monitors in the correct locations, but also to allow campers to model the staff members' behavior. Finally, the camp counselor assigned to each participant, was asked to encourage compliance from each camper. Of the 145 accelerometers distributed (29 participants over 5 days), only one device was lost.

Accelerometers were distributed each morning, prior to the start of the camp activities. Along with accelerometers, a wear-location and wear-time log was distributed to the specific counselor assigned to each participant. Participants were given autonomy to determine wear location (hip, wrist, or ankle). Participants were informed that they could remove and re-attach the devices at any time. Appropriate wear locations were thoroughly explained to counselors. Wrist-worn accelerometers were located proximal to the ulnar styloid process. Ankle-worn accelerometers were located on the lateral aspect of the ankle, proximal to the lateral malleolus. Hip-worn accelerometers were placed above the right iliac crest on the axillary line. A picture of each wear location was also provided to each counselor. Counselors were instructed to record wear time on the log provided. In addition, wear location was also recorded. Participants were asked to wear the devices for the duration of the day, excluding water-related activities. At the end of each day, accelerometers were collected, data were downloaded, and then the devices were initialized and redistributed the next morning. The majority of participants (22 out of 29 enrolled) chose to wear the accelerometer on their wrist and the remaining participants chose to wear the device on their ankle. For the purpose of research, the wrist site is the most commonly used site and is able to record upper limb movement in addition to ambulation. This site has also been shown to promote better wear compliance and be perceived as less burdensome compared to the hip location [27,28] National studies such as U.S. NHANES and the UK Biobank Study utilize the wrist location [24,25]. The ankle only records ambulatory activity, which may limit measurement of true activity levels especially in individuals with physical disabilities, who may be limited in their ability to perform ambulatory activities. The wrist placement is also the most common site for wearable technology (Apple Watch, Fitbit, etc.) and may facilitate the use of these devices to monitor activity levels in individuals with developmental disabilities. Therefore, the authors chose to only use the participants with wrist data ($n$ = 22) in analysis as these data provided the most comprehensive measurement of physical activity in the sample. Although there were no differences in accelerometer data between the wrist and ankle ($p > 0.05$), we chose to only use data from participants who wore the accelerometer on their wrists. For the reasons described above and on account of the diverse nature of the camp activities, using the wrist location allows for consistent data measurement of upper and lower body movement among all participants.

#### 2.2.1. Accelerometry

Physical activity data were collected using ActiGraph GT3X and GT3X+ (ActiGraph, Pensacola, FL, USA) accelerometers. Accelerometers measure body movements in terms of acceleration of three axes (vertical, medio-lateral, and antero-posterior) [29]. The data from these axes were exported into the ActiGraph software (ActiLife, version 6.13.4; Pensacola, FL, USA) to calculate vector magnitude (VM). VM was determined by taking the square root of the sum of squares of each axis. The device was initialized to collect data in 1-s epoch lengths. VM was chosen to represent overall movement of the participants since accelerometer cut-points for intensity level (sedentary, light, moderate, and vigorous) have

not been established for youth with disabilities and VM provides information on activity in all axes.

VM data were converted to activity intensity levels using the cut points developed by Crouter and colleagues [28]. These cut-points were chosen since they were utilized in youth close to the age range of those in the current study and have been utilized in youth physical activity surveillance research [30]. Additionally, these cut-points were chosen since there are currently no published wrist cut-points developed on youth with developmental disabilities. Although these cut points are not ideal because they were not developed on youth with developmental disabilities, they allowed the authors to provide information regarding the general intensity level of the activities completed during the camp. Previous studies successfully utilized accelerometers in youth with disabilities (Cerebral Palsy, Muscular Dystrophy) and included wear on both the wrist as well as the hip [31,32]. For the purpose of the current study, compliance with accelerometer wear was defined as wearing the monitor during the specific camp activities. Wear-time validation was computed using the Choi [33], provided by the ActiGraph software and was compared to the wear time recorded by the counselor.

### 2.2.2. Data Analysis

Data were analyzed using SPSS version 25 (IBM, Armonk, NY, USA). All values are presented as mean $\pm$ standard deviation (SD). Any activity where the individual wore the accelerometer for the entire duration of the activity was included in the analysis. On average participants complied with wearing the accelerometer during 81.4% of the activities and data were available for 82.1% of the sample for all of the activities. Overall, this indicates that participants were compliant and wore the accelerometers during the majority of the activities during camp. Accelerometer data were not normally distributed; therefore, Kruskal–Wallis tests were utilized to determine differences in VM among activity categories and individual activities. Post hoc pairwise comparisons were run to determine where differences existed amongst activity categories and activities and a Bonferroni correction was applied. To provide contextual information about the activity, Crouter et al. cut-points were used to classify the average intensity level of each activity [28]. Hierarchical linear models were used to demonstrate the time course change in VM with time over the course of an activity, as the level one variable, the individual participant being the level two variable, and the predictor variable of activity type (ropes, outdoor adventure, horses, music and movement, come on down, sports and games, theatre, cabin challenge, arts, cooking, mad science, yoga, free period). Significance was set at $p < 0.05$ a priori.

### 3. Results

Mean VM data for activity categories and each activity are presented in Table 4. There were differences in activity levels amongst the activity categories [H(4) = 32.57; $p < 0.001$]. Activity during the Free Period category was significantly greater than the Gross/Functional ($p = 0.002$), and Fine/Constructive ($p < 0.05$) categories. Sociodramatic activity was greater than Gross/Functional ($p < 0.001$), but less than Fine/Constructive activity ($p = 0.001$). There were also significant differences in VM among the different activities [H(12) = 311.21, $p < 0.001$]. There was more activity ($p < 0.005$; except where noted) during unstructured free time, horses, sports and games, outdoor adventure, and cabin challenge compared to yoga, music, mad science, come on down, and ropes (ropes and horses compared to free play; $p < 0.05$). Cooking resulted in higher activity levels ($p < 0.001$; except where noted) compared to yoga, music, mad science, come on down, arts, ropes, and theatre ($p = 0.009$). On average, the participants engaged in mostly sedentary and light intensity physical activity during the camp activities with minimal time spent in MVPA (Table 4).

There were main effects for time, $F(1, 2349.65) = 54.26$, $p < 0.001$ and activity, $F(12, 5835.62) = 7.17$, $p < 0.001$, and a significant interaction between time and activity, $F(12,$

5841.43) = 5.40, *p* < 0.001. Figure 1 represents the average time course change of VM for each activity and they are grouped by activity category.

**Table 4.** Vector magnitude (mean and SD) and activity intensity for activity categories and activities.

| Activity Category (Activity) | Vector Magnitude (Counts per Minute) | | | Activity Intensity (%; Min during the 41-min Activity Session) | | |
|---|---|---|---|---|---|---|
| | *n* | Mean (SD) | Median (IQR) | Sedentary | Light | Moderate |
| **Gross/Functional** | **3690** | **2807.2 (2383.7)** | **2363.6 (1145.2–3831.4)** | | | |
| Ropes | 779 | 2698.5 (2130.7) | 2239.4 (1192.0–3691.3) | 75.6 (31 min) | 19.5 (8 min) | 4.9 (2 min) |
| Horses | 738 | 3409.2 (2458.5) | 3073.8 (1719.6–4444.1) | 58.5 (24 min) | 41.5 (17 min) | 0 (0 min) |
| Outdoor Adventure | 779 | 3234.6 (2700.4) | 2710.9 (1584.0–4168.7) | 53.78 (22 min) | 46.3 (19 min) | 0 (0 min) |
| Music & Movement | 697 | 2208.2 (1880.1) | 1813.7 (865.8–3128.6) | 92.7 (38 min) | 7.3 (3 min) | 0 (0 min) |
| Yoga | 697 | 2412.2 (2416.1) | 1872.4 (800.1–3326.4) | 87.8 (36 min) | 12.2 (5 min) | 0 (0 min) |
| **Game** | **1558** | **2938.9 (2461.4)** | **2481.9 (1198.2–4050.9)** | | | |
| Come on Down | 738 | 2527.6 (2074.9) | 2119.5 (998.4–3555.1) | 90.2 (37 min) | 9.8 (4 min) | 0 (0 min) |
| Sports & Games | 820 | 3309.1 (2715.5) | 2752.8 (1412.8–4498.11) | 58.5 (24 min) | 41.5 (17 min) | 0 (0 min) |
| **Sociodramatic** | **1599** | **3155.0 (2580.6)** | **2650.4 (1446.3–4304.1)** | | | |
| Theatre | 861 | 3080.6 (2715.5) | 2538.3 (1373.9–4176.1) | 68.3 (28 min) | 29.3 (12 min) | 2.4 (1 min) |
| Cabin Challenge | 738 | 3241.9 (2412.7) | 2882.4 (1535.7–4485.0) | 63.4 (26 min) | 36.6 (15 min) | 0 (0 min) |
| **Fine/Constructive** | **2127** | **3225.8 (3393.1)** | **2295.4 (1036.4–4157.6)** | | | |
| Art | 738 | 2876.0 (2742.8) | 2183.0 (1339.0–3351.0) | 80.5 (33 min) | 19.5 (8 min) | 0 (0 min) |
| Cooking | 697 | 4379.8 (4403.4) | 3281.0 (1046.5–6371.9) | 2.4 (1 min) | 65.9 (27 min) | 31.7 (13 min) |
| Mad Science | 692 | 2436.7 (2402.3) | 1887.0(669.3–3425.4) | 87.8 (36 min) | 12.2 (5 min) | 0 (0 min) |
| **Free Period** | **697** | **3610.2 (3518.8)** | **2914.2 (933.9–5334.2)** | | | |
| Unstructured Free Time | 697 | 3610.2 (3518.8) | 2914.2 (933.9–5334.2) | 31.7 (13 min) | 68.3 (28 min) | 0 (0 min) |

SD = Standard Deviation; IQR = Interquartile Range.

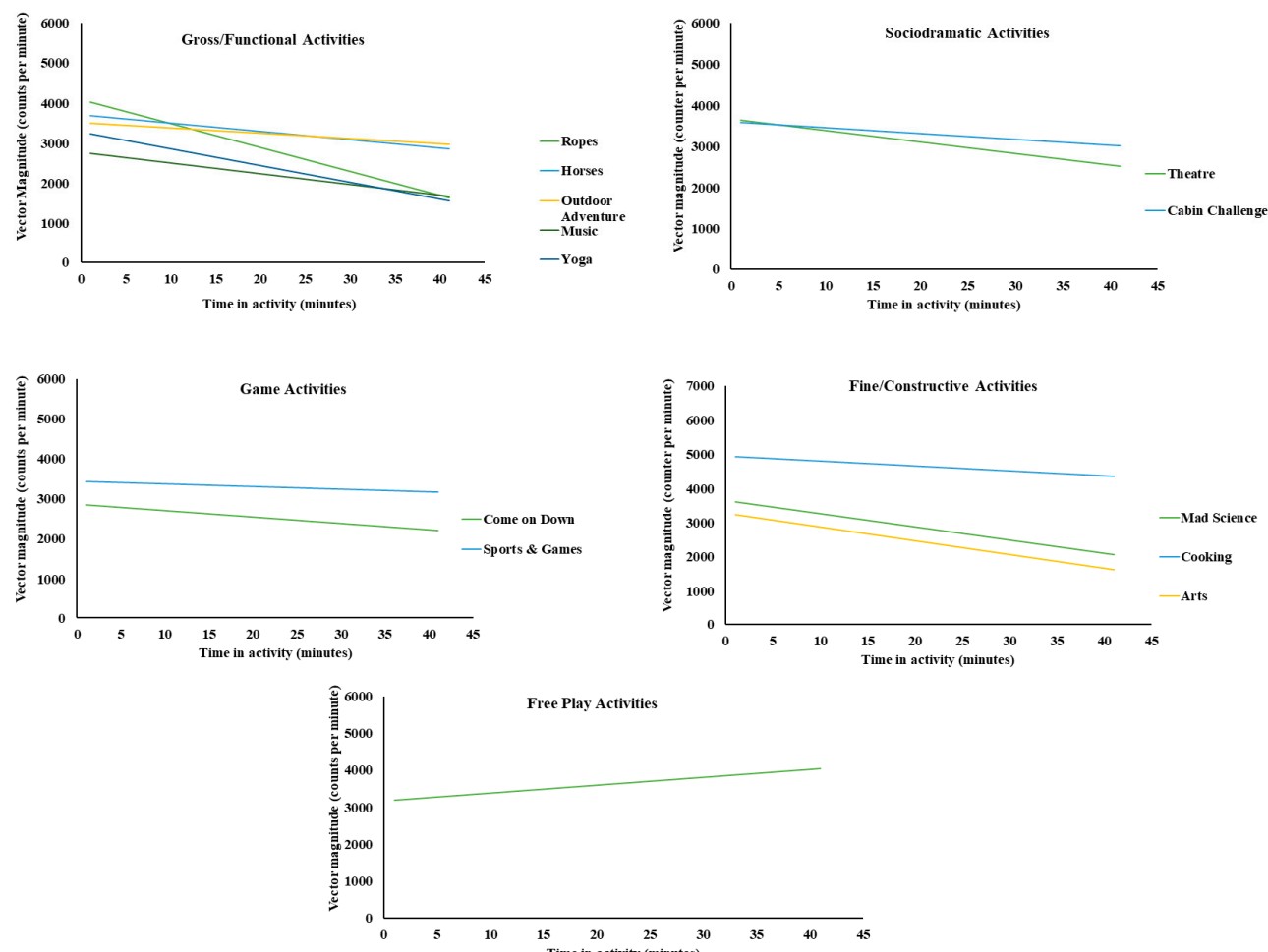

**Figure 1.** Representation of the time course activity during different classifications of activity.

## 4. Discussion

Current evidence suggests that youth with developmental disabilities are at an increased risk for physical inactivity and its related consequences. Through the identification of activities that can be used to reduce sedentary time, strategies can be developed to increase overall levels of physical activity. The purpose of this study was to measure the physical activity levels of youth with developmental disabilities attending a therapeutic camp through the use of wearable accelerometer technology. It appears in this investigation that the physical activity levels (VM) during an assortment of activities varied by activity category as well as by activity. This finding was supported by growth curve analyses that revealed the physical activity levels varied throughout the duration of each activity based on activity type and category. The majority of the activities saw a decline in VM over the activity time period. Cooking had the highest amount of activity over the time period, and it appears that both cooking and sports and games remained relatively consistent over the activity period. The free play period was the only activity that saw a linear increase over duration of the activity period. During all camp activities, the participants engaged in mostly light intensity activity, limited moderate intensity activity (~3% of the time), and the remainder of the time in the activity spent being sedentary. Finally, the participants spent the most time in light intensity activity when participating in the cooking and free play activities.

Overall, the average activity level differed by activity category, as well as by activity. Accelerometry has been used in a variety of studies to explore the physical activity levels of children without disabilities during horseback riding [34], sociodramatic play [35], dancing activities [36], arts and crafts [37], and during recreational sports [38]. Additionally, a recent study explored the activity levels of children with autism spectrum disorder using wrist-worn accelerometry [39]. However, to the authors knowledge, no studies have provided information on the activity levels of all the activities explored in the current investigation. Camps in general offer an opportunity for youth with and without developmental disabilities to increase physical activity levels through a variety of activities [40,41]. Consistent with previous research, the results of the current investigation suggest that physical activity can be accumulated regardless of the camp activity.

According to Leung and colleagues, it is recommended that accelerometer data be reported as raw outputs or as activity counts over a specific time period (i.e., counts per minutes) since no cut-points for the wrist have specifically been developed for youth with developmental disabilities [42]. Additionally, this would allow for comparisons between studies to be made. VM data in this study revealed differences in activity categories. Unexpectedly, participation in the gross motor/functional activities had the lowest level of activity counts. These types of activities use large muscles and typically are continuous in nature. However, the gross motor/functional activities had the largest decrease in activity during the activity period (Figure 1). This decrease could be due to a number of factors that include a lack of stamina to complete the activities, lack of skill, increased time devoted to familiarization with the activity, or boredom during the activity. Free play had one of the highest VM of all activity categories and on average activity counts increased throughout the duration of the free play (Figure 1). During free play, it may take time for the individual to decide on a chosen activity leading to low activity counts. Initiation and participation in the activity for the remainder of the activity period may result in the gradual increase in activity levels during free play.

In the current study, cut-points from Crouter et al. were used in order to provide some information regarding the general intensity levels of all of the individual camp activities as well as to put the VM counts per minute into context [28]. The participants in this study spent a large percentage of time being sedentary during the camp activities, with the exception of cooking, sports and games, and free play. However, many of the activities allowed the participants the opportunity to engage in some light intensity physical activity. Although few of the camp activities evaluated elicited MVPA in the study participants, they offer the opportunity to engage in light intensity activity and to reduce sedentary

behaviors. Higher levels of sedentary behavior are associated with lower levels of physical activity and more screen time, which have a deleterious impact on the health of children and adolescents [43]. Light intensity activity does have a lower cardiovascular demand and may be more comfortable for youth, yet light activity is still meaningful from a public health perspective [44]. Additionally, emerging research has shown a correlation between light physical activity and reduced overall disease risk [43]. The WHO has identified the importance of light activity as it is more likely that individuals will engage in light intensity activity for longer periods of time compared to MVPA [7]. While the participants accumulated very little MVPA to meet recommendations during camp activities, the activities did allow for light intensity activity and reduction in sedentary time, as well as the opportunity to learn new skills and engage in group activities (structured and unstructured) with their peers. Since activity levels tend to be low in this population, any movement regardless of intensity may be beneficial.

The camp environment offers a unique opportunity for health care professionals, parents, and therapists to address other skills (e.g., cognition, socialization) while increasing physical activity. To increase physical activity adherence, it is recommended that programs be designed to be diverse and autonomous [45]. The current investigation supports this notion, suggesting that physical activity levels can be increased with a variety of activities. Therefore, it may be pertinent to focus more on discovering activities that intrinsically motivate youth as opposed to those that are traditionally thought to increase physical activity (e.g., recreational sports).

As previously stated, of all the activities analyzed in the investigation, free play appeared to be the only activity to linearly increase with time. Additionally, the free play category activity was significantly higher than the Gross/Functional and Fine/Constructive categories. The free play session allowed campers to choose an activity they wanted to participate in with support and limited restrictions from the camp staff. The authors postulate that the ability to participate in activities of preference for the camper reinforces the notion that intrinsic motivation is critical to increasing physical activity levels [46]. Free play is important for children with developmental disabilities for a variety of reasons including improving social interaction [47], improving the ability to learn in an inclusive setting [48], and providing a foundation to develop leisure skills [49]. In a typical day-to-day situation, this free play can be compared to recess while at school. Since the play opportunities outside of the school setting remain scarce [50], it is recommended that children with developmental disabilities accumulate physical activity within the school setting [51]. Due to the freedom associated with playing during recess, this may be an ideal opportunity for youth with developmental disabilities to participate in physical activity. Although the benefits of recess are well known [52], current evidence suggests that, during this time, those with developmental disabilities do not accumulate enough physical activity [51,53]. Since free play activity gradually increases with time in the current study, opportunities for longer periods of free play should be available to encourage physical activity. Future research should focus on developing strategies to increase opportunities to engage in free play in a variety setting outside of the camp environment, especially within the school setting.

When given the opportunity for free play, the level of physical activity typically does not remain consistent throughout the free play period. Research in pre-school children conducted by Pate et al. showed varying patterns of activity during outdoor, free play time [54]. Boys showed a gradual decline throughout the period while the girls' activity was episodic, with variations in active and sedentary periods. Overall, Pate et al. found that activity is typically highest at the beginning of the play period and gradually decreases [54]. This pattern of activity is similar to that found in the current study. With the exception of cooking, sports and games, and free play, there was a decline in activity level throughout each session. This may be indicative of a loss in interest with the activity over time or the inability to sustain the activity. During cooking and sports and games, activity remained relatively consistent and during the free play session, there was a gradual increase in activity

over the 41-min time period. During the free play, it may take time for the individual to decide what activity to engage in and then they may increase physical activity levels as they participate in their chosen free play activity. These findings indicate that the time period of the activity should be considered when creating programs for youth with developmental disabilities.

The camp that served as the setting for the current study had the facilities and all of the equipment necessary to implement a variety of structured activities. During free play at camp, the participants were not limited to a specific area and had access to equipment to use during this activity period. Environments designed to promote activity through fixed elements (i.e., soccer fields, playgrounds) and portable objects (i.e., soccer balls, hula hoops, etc.) allow youth to self-determine their own activity. This autonomy may lead to a more pleasurable and sustainable physical activity opportunity. This is consistent with current literature, in which these types of structured environments (e.g., school based physical activity) offer an opportunity for youth with developmental disabilities to increase their physical activity levels [16,55]. It appears that, in this investigation, offering a self-determined environment to engage in free play while also having portable objects and fixed structures, may allow children with developmental disabilities to autonomously engage in physical activity. Therefore, it is important that future interventions consider a supportive and structured environment, that also provides adequate supplies that allow for freely chosen and self-determine desired activities.

Some activities such as cooking, or arts may be classified as sedentary if using accelerometer data from the hip due to lack of ambulation during the activities. However, when worn on the wrist the accelerometer records activity despite an absence of ambulation. These findings are consistent with previous literature exploring activity of children without disabilities in a free-living environment [54]. Accelerometers worn on the wrist are sensitive to extraneous movements of the hands [56–58]. Since cooking is an activity that requires a lot of activity utilizing the hands, it is possible that these movements resulted in higher VM compared to other activities that required more gross motor movements and less arm gesticulation and variability. A youth physical activity compendium has recently been developed by the National Collaborative on Childhood Obesity Research, but cooking is not an activity included in this document [59]. Since cooking has been shown to be enjoyable and sustainable [60], while expending low amounts of energy, it may be a possible pragmatic activity for youth with developmental disabilities. Thus, while cooking involves continuous fine motor hand-gestures, yet requires relatively low activity intensity, this may explain why the participants were able to maintain a relatively consistent VM throughout the duration of the activity.

There are several strengths to the current study. The use of objective monitors provides a more accurate assessment of physical activity levels since it does not rely on recall bias associated with questionnaires or the subjective bias of direct observation. The environment where the study took place enabled close, continuous monitoring of the participants' compliance with wearing the accelerometer. Due to the 1-to-1 ratio of camper to counselor, actual wear time data were recorded for each participant. The camp provided a unique backdrop that allowed for the quantification of a wide variety of autonomous activities that encompassed diverse activity categories. The current investigation is not without limitations. Since the activities were performed in a 1-to-1 camp setting, counselors and camp staff were consistently able to motivate participants to remain engaged in activities. Due to the fact that all participants were diagnosed with a variety of developmental disabilities, data generated from the accelerometer may be impacted by gait irregularities. However, accelerometers have been used in these populations previously and are currently the most feasible way to measure physical activity in youth with developmental disabilities. Therefore, it is difficult to generalize these results to the day-to-day life of children with developmental disabilities. Future studies should explore this relationship in a more typical environment. In addition, future research should explore compliance and physical activity levels with a more homogenous population. The sample size was relatively small which

may have had an impact on the study results. Finally, since the counselors were responsible for recording the compliance of the participants, it is possible that there were errors in the recording of this information and implementation of these devices.

There are several important implications regarding the current study. The use of objective monitors that are able to discriminate among varying intensity levels provides valuable data on specific activities that are typical in therapeutic settings. According to the American College of Sports Medicine Worldwide Survey Fitness Trends 2021, wearable technology was the second most commonly identified trend by fitness professionals. This trend has remained a popular modality (top 3 every year) since being introduced in 2016. This suggests that wearable technology (i.e., wearables) is recognized as a popular positive tool in the fitness industry [61]. However, there is a lack of data in youth with developmental disabilities regarding physical activity participation using objectively monitored physical activity levels. By presenting accelerometer data, this may provide information for future investigations to compare to, with the intention to gain a more robust understanding of the physical activity patterns of youth with developmental disabilities. This is important as the WHO has identified this evidence gap and has determined that observational studies, randomized control trials, and mixed method studies need to be conducted in youth with developmental disabilities in order to clearly determine the relationships between physical activity, sedentary behaviors, and health and wellness outcomes [43]. The current study provides observational data that can be used as a starting point to develop future studies and to determine activities that may be useful to increase movement in this population.

## 5. Conclusions

Improving the physical activity levels of youth with developmental disabilities remains the uppermost priority of researchers and practitioners and this research supports the use of wearable technology such as accelerometers to measure physical activity in youth with developmental disabilities. During the study, the youth were able to accumulate some light intensity activity during all of the camp activities. Thus, the camp setting is a way to expose youth to new activity opportunities as well as to encourage physical activity. The results suggest that a variety of activities may be used to increase physical activity, which may offer the opportunity to address other skills such as social, emotional, and behavioral skills. Finally, unstructured free play appears to linearly increase as time increases. It appears that when given the opportunity to play freely, youth with developmental disabilities will gradually increase their physical activity levels. This further supports the notion that youth with developmental disabilities should engage in activities that are intrinsically motivating.

**Author Contributions:** Conceptualization, V.G.N., A.J.W. and D.P.C.; Methodology, V.G.N., A.J.W. and D.P.C.; Investigation, V.G.N. and T.J.K.; Data Curation, V.G.N. and T.J.K.; Formal Analysis, D.P.C.; Writing—Original Draft Preparation, V.G.N., A.J.W. and D.P.C.; Writing—Review and Editing, V.G.N., A.J.W., T.J.K. and D.P.C. All authors have read and agreed to the published version of the manuscript.

**Funding:** This research received no external funding.

**Institutional Review Board Statement:** The study was conducted according to the guidelines of the Declaration of Helsinki and approved by the Institutional Review Board of The University of Tennessee, Knoxville (UTK IRB-18-04315-XP; 3 May 2018).

**Informed Consent Statement:** Informed consent was obtained from all subjects involved in the study.

**Data Availability Statement:** For more information regarding data availability, please email Dawn P. Coe, dcoe@utk.edu.

**Conflicts of Interest:** The authors declare no conflict of interest.

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
