# Peer review of "Physical Activity Levels during Therapeutic Camp Activities in Youth with Disabilities in the United States"

_disabilities, doi:10.3390/disabilities2040053_

Round 1

Reviewer 1 Report

Comments and Suggestions for Authors

Please find my comments in the file attached.

Reviewer 2 Report

Comments and Suggestions for Authors

I would like to thank you for the opportunity since I feel very fortunate to be able to review this article and I would like to congratulate the authors for this work. For me, as a physical educator specialized in outdoor physical activity, this topic is very important and has a lot of value. I detail my suggestions below and at the end my consideration.

This manuscript investigated physical activity levels during therapeutic camp activities in youth with disabilities.

Title: The title is concrete, representative and indicative of the problem investigated in the manuscript. As a suggestion, the title should provide information about where the research was conducted.

Abstract: The abstract is clear and complies with the general rules for writing a good abstract. However, I would like to see a better description of the sample, indicating the context. I would also improve the description of the camp (duration (5 days with overnight stay...), environment...) This is the most important section of the document since it will be read many more times than even the manuscript itself, so it needs the most attention. A brief note on the importance of the research is an excellent end to a high level summary.

Introduction

As I mentioned, I find this research extremely important in contributing to the field of Physical Activity and Disability. I do not disagree with the authors' justifications and read many very good and current arguments.  However, it is suggested to provide more information about the conditions that must be met to be considered "therapeutic camp", what conditions must these activities meet to be considered therapeutic? is it the same to carry out a camp with overnight stay than without overnight stay?

It is suggested to the authors that, based on the stated objective, they highlight the research questions that will help to conduct the research and the discussion based on the findings, in which the study variables, the study population, and the expected result appear.

Material and method.

Participants. Inclusion and exclusion criteria should be indicated.

Statistical analysis. It is reported that the assumption of normality is not fulfilled. The choice of tests should be justified. What were the cut-off points for classifying the level of intensity of the activities?

Results: 

The results are displayed correctly and are easy to read and simple for a scholar not used to quantitative methodology. However, as this is a non-parametric statistic, it is also suggested to present the data as median and IQR (interquartile range).

Discussion: It seems to me that a great job has been done in comparing the findings with other studies. Congratulations. It is suggested to include a section on practical and theoretical implications to evaluate the scope of the research.

Conclusions: They are clear and provide an answer to the stated objectives. 

Round 2

Reviewer 1 Report

Comments and Suggestions for Authors

The authors have made sufficient changes to the manuscript that clarify concerns and highlight the study’s contributions to the literature. The change from MVPA focus to light PA and reduction of sedentary time strengthens the arguments made by the authors. The article is well written. The authors should review for some formatting errors (e.g., table length).   

Reviewer 2 Report

Comments and Suggestions for Authors

I thank the authors for having taken my comments into account.  I consider that the article has been sufficiently improved and can be published in this journal. Congratulations